# Elementary Students’ Knowledge Development during the Implementation of “After School Exercise” Program

**DOI:** 10.3390/children8030248

**Published:** 2021-03-23

**Authors:** Ioannis Syrmpas, Marios Goudas

**Affiliations:** Department of Physical Education and Sport Science, Lab of Exercise Psychology & Quality of Life, University of Thessaly, Karyes, 42100 Trikala, Greece; mgoudas@pe.uth.gr

**Keywords:** knowledge, goal setting, goal plan, physical education, benefits of physical activity

## Abstract

Physical education should focus not only on students’ motor and emotional development but also on their cognitive development. The purpose of the present study was to examine whether elementary students’ health-related knowledge and physical activity-related goal setting increased after they participated in a program. The program aimed at promoting after school physical activity among students. Participants were 244 fifth- and sixth-grade students (116 boys and 128 girls). Students’ knowledge was examined by means of a multiple-choice test, cognitive assignments and a retrospective pre–post questionnaire. The results from the three measures indicated that students enhanced their knowledge both regarding the health benefits of physical activity (PA) and effective goal setting. Hence, it can be argued that the program was effective in promoting students’ skills and knowledge related to PA.

## 1. Introduction

Physical activity (PA) may lead to an array of benefits for people’s health [1,2]. A significant number of studies indicated that regular participation in PA could be beneficial for the immune system, and it also protects the body against cardiovascular diseases and aids the prevention, treatment and control of hypertension, diabetes, obesity, osteoporosis and depression [3,4]. The World Health Organization (WHO) [5,6], taking into consideration the aforementioned findings, recommends that children should be involved in moderate to vigorous PA at least one hour per day.

Students’ lifelong engagement in PA has been adopted as one of the physical education (PE) goals [7]. Similarly, PE in Greece focused on the improvement of students’ fitness and health through their motor development and the adoption of a physically active lifestyle [8]. Ηowever, the findings of a study [9] articulated that children do not meet the WHO’s recommendation [5]. Further, the WHO [10] reported that only 12% of children in Greece over 13 and 14% over 15 meet the WHO’s recommendation. The WHO [11], in order to address the decline in youth PA participation, proposes that PE should play a pivotal role in promoting PA. The time allocated for PE is limited (45–120 min per week) and thus is not sufficient to meet the WHO’s recommendation [12]. Interestingly, the findings of a study revealed that children 8 to 12 years old were not aware of recommended levels of frequency, intensity and duration of PA [13]. Therefore, an action plan is needed so that PE teachers can adopt the appropriate strategies to promote students’ participation in after school PA [11]. One such strategy may be the enhancement of students’ knowledge regarding the health benefits of PA and the development of skills related to PA behavior.

### 1.1. Learning in the PE Context 

Learning in the PE context refers to the development of students’ cognitive, psychomotor and emotional skills [14]. Penney and Jess [15] stressed the need for the development of a multidimensional PE curriculum, aiming at helping students to be lifelong learners by applying the knowledge and skills they learned in PE throughout their life. Additionally, Gallahue and Donnelly [16] argued that learning in the PE context is a complex process because it comprises three independent dimensions: cognitive, motor and affective. They also stressed that cognitive learning includes a wide variety of PE concepts, some of which are common to other academic domains (anatomy, physiology, etc.). Arguably, Ennis [17] articulated that cognitive learning should take a central role in the PE context. 

Vosniadou [18] argued that learning is domain-specific. She also stressed that learning is a gradual, slow and longitudinal process. Learners’ prior knowledge plays an important role in their attempt to understand and give rational explanations about every new piece of information or problem that they have to deal with. Learners’ knowledge is enriched or radically reconstructed under the influence of personal, social, cultural and contextual factors [19,20]. Vosniadou [21] argued that when the provided knowledge is consistent and meaningful to students’ prior knowledge, then learning is easy to occur. 

Researchers [22] pointed out that learning is more effective when PE provides students with opportunities to be actively engaged in the learning process by converting the theory into practice and applying the new knowledge to their daily reality. Similarly, PE theorists [23,24] argued that the integration of physical and cognitive tasks in PE is an effective strategy to facilitate students’ cognitive learning. 

In the last two decades, a number of fitness education curricula have been developed, e.g., [25], aimed at helping students to learn concepts of fitness through their participation in PA. Similarly, curriculum developers emphasized the cognitive dimension of learning by integrating health-related concepts in PE [26,27]. Ennis [17] characterized the aforementioned curricula as concept-based PE curricula because they include knowledge about PA. 

### 1.2. Health-Related Knowledge

Keating and her colleagues [28] defined health-related knowledge as the “knowledge about individuals’ ability to perform PA and protect themselves from chronic disease” (p. 335). Keating [29] argued that helping students to develop health-related knowledge could lead them to adopt healthy PA behaviors. Similarly, Wang and Chen [30] suggested that students’ motivation and health-related knowledge could act as catalysts on students’ participation in after school PA. Interestingly, the findings of a study revealed that students’ health-related knowledge was related to their participation in PA [31]. Further, the implementation of concept-based PE curricula, e.g., [30,32,33], increased students’ health-related knowledge and promoted students’ participation in after school PA. However, the findings of previous studies revealed that students at elementary school [34,35], middle school [36] and high school [28] have limited knowledge of health-related concepts. 

Researchers aiming at enhancing students’ health-related PA knowledge have included cognitive assignments in respective programs and curricula, e.g., [25,27]. The results of respective studies showed that the inclusion of cognitive assignments in the PE curriculum was beneficial regarding students’ knowledge [30,37]. Interestingly, the findings of a study [22] showed that the incorporation of cognitive assignments in the PE lesson increased students’ knowledge even when answering incorrectly in comparison to skipping the assignments.

### 1.3. Goal Setting in the Context of Sport, PA and PE

In the context of sport, PA and PE, three types of goals have been identified: process, performance and outcome. Process goals refer to the strategies or behaviors a person adopts during the task. Performance goals, on the other hand, refer to goals’ accomplishment or failure and emphasize personal improvement based on personal criteria [38,39]. An example of a performance goal is a person’s goal to improve their performance in push-ups from 10 to 13 repetitions after one month. Outcome goals ultimately emphasize the result of performance involving social comparison [39]. For example, a person’s goal to outperform their classmates in push-ups test is considered an outcome goal. In the context of the present study, process goals refer to the behaviors students adopt in order to achieve a performance goal and resemble the plan of achieving a performance goal (e.g., doing three sets of ten push-ups on a Thursday afternoon after reading).

The adoption of goal-setting theory [40] can lead to positive results in many contexts. For example, Epton and her colleagues [41] pointed out that the goal-setting theory was effective in adopting or improving health-related behaviors, sport skills and cognitive learning. Similarly, the findings of a cross-sectional study [42] showed that the implementation of goal-setting theory has a positive impact on health-related behaviors such as weight loss and body mass index, reduction in food and beverage consumption and increase in energy expenditure. In order to articulate characteristics of effective goal setting, Doran [43] and Hersey and Blanchard [44] formulated the acronym S.M.A.R.T. to outline the characteristics of effective goal setting (specific, measurable, activity-related, realistic and time-based).

PE is the ideal environment for teaching strategies such as goal setting, self-assessment and self-monitoring [45]. Goal-setting theory has been applied in the PE context with positive results. For example, it has positive effects on students’ development of social responsibility [46]. Additionally, the implementation of goal setting in combination with self-monitoring led to improved student performance in an aerobic capacity test [47]. Three consecutive studies have shown that applying goal setting in combination with problem-solving strategies and positive thinking led students to improve volleyball skills (e.g., finger passes) [48], basic physical qualities (e.g., flexibility, strength) [49] and basketball skills (e.g., dribbling) [50]. Furthermore, the above interventions were effective in improving students’ knowledge regarding goal setting. Finally, the findings of studies [51,52] showed that applying goal-setting theory can effectively increase students’ participation in PE. Although the previous studies indicated that implementation of goal setting can lead to positive outcomes, Baghurst, Tapps and Kensinger [53] stressed that the incorrect use of goal-setting theory could decrease students’ performance in the PE context.

Studies that have applied goal setting in the PE context [48,49] examined students’ knowledge of goal setting by means of a multiple-choice test and reported that participants in the intervention group scored higher than participants in the control group. However, these studies did not use cognitive assignment to examine students’ knowledge of goal setting. Other studies [30,33,37] examined students’ PA health-related knowledge (including goal-setting knowledge) by using only cognitive assignments. Finally, some studies [22,28,34,35,36] examined students’ health-related knowledge through knowledge tests. The current study adopted a more comprehensive strategy for assessing students’ knowledge employing a multiple-choice test, cognitive assignments and a retrospective questionnaire. The study aimed at examining whether students who participated in the “After School Exercise” program increased their knowledge regarding goal setting and the health benefits of PA.

## 2. Materials and Methods

### 2.1. Participants

An invitation to 60 PE teachers who had connections with the authors’ Department of PE was forwarded. Thirty-one of them accepted the invitation and delivered the “After School Exercise” program. They taught in schools located in different geographical areas of Greece. Researchers aimed at adopting a two-group repeated-measures reverse-treatment quasi-experimental design. However, due to the lockdown measures authorities imposed in response to the outbreak of coronavirus, the second phase of the deigned study was not conducted. Consequently, 268 students participated in the study, and 244 of them (boys = 116, 47.5%, girls = 128, 52.5%) provided complete data. The study was implemented with the approval of the Ministry of Education and Ethics Committee of the authors’ university. Parents/guardians’ consent and students’ assent forms were secured before data collection.

### 2.2. The “After School Exercise” Program

The aim of the program was to promote the participation of fifth and sixth graders in after school PA. The development of the program was based on relevant school-based interventions and the theoretical background of the sociocognitive model of self-regulated learning. A teacher book, a student book and a student workbook were developed. A series of 16 sessions were designed, and respective ready-made teaching plans were included in the teacher book alongside with suggestions for effective delivery of the program. Both the teacher and the student book include information on the (a) aims of the program, (b) health benefits of the PA, (c) recommended levels of PA, (d) characteristics and types of PA, (e) components of physical fitness, (f) measurement of heartbeat and estimation of PA intensity and (g) benefits of goal setting coupled with examples of effective goal setting. Each teaching plan includes an introductory, a main and a final part. During the introduction, the PE teacher can inform students about the aim of the session and introduce them to the new information. The main body includes activities in which students can apply the information that they have just received. During the final part, the PE teacher should discuss with students and prompt them to apply the knowledge and activities that they received during the lesson in their daily life along with their parent(s)/guardian(s), sibling(s) and/or friends. For example, during the fifth session, the PE teacher introduces students to goal-setting theory. More specifically, they inform students about the importance and usefulness of setting goals. They also introduce them to S.M.A.R.T. principles. During the main part of the session, the PE teacher must ask students to evaluate the strength of their abdominal and dorsal muscles by choosing an activity commensurate with their skills. Then, based on their performance, students should set a goal for enhancing their performance within the next month. At the end of the session, the PE teacher should discuss with students the implementation of the goal-setting theory in another context. Finally, they must prompt them to apply the goal-setting theory along with their parent(s)/guardian(s), sibling(s) and friends. Overall, a combination of physical and cognitive skills is promoted in order to motivate students to increase their PA. A detailed presentation of the “After School Exercise” program can be found in [54].

The student workbook includes a multiple-choice test, questionnaires and cognitive assignments. More specifically, the students’ workbook includes three assignments in which students should design three PA plans based on information they received during the implementation of the program. For example, in the seventh session, the following assignment is included: “Please write in the table below what day, how much time and what activities you will do next week to improve the strength of their arms, abdominal and dorsal muscles”.

### 2.3. Instruments

Multiple-choice test. A multiple-choice test was developed and then evaluated by a panel of two content experts through a review and revision process and confirmation of its content validity. Additionally, the test was piloted with fifth- and sixth-grade students. The initial version of the multiple-choice test consisted of 24 items. PE teachers from the pilot study reported that students spent more than 20 min filling in the multiple-choice test. Researchers, in order to be aligned with the Ministry’s guidelines, according to which the delivery of the program should not affect students’ PA, decided to choose nine (the most representative) items of the initial version of the multiple-choice test. Additionally, researchers, aiming to avoid performing nine paired sample *t*-tests, decided to group the nine items into the three following categories: (a) knowledge related to health benefits of PA (three questions), (b) knowledge related to goal setting (three questions) and (c) knowledge related to choose activities suitable for strengthening specific body parts. However, each item depicts a different aspect of the category in which it has been included. For, example, the stem “Exercise is beneficial for your health when you participate …” was used to capture students’ knowledge of recommended PA levels for children. On the other hand, the stem “PA is considered …” was used to examine students’ awareness of the variety of types of PA. Finally, the stem “Your participation in PA helps you…” was used to examine students’ awareness of the benefits of PA. It can be argued that these items depict different aspects of knowledge related to the health benefits of PA, and thus, it is not possible for the construct validity and reliability of the items to be established. Students could answer each question by choosing one of a set of four alternative answers. Each correct answer was coded as one (1) and each false answer as zero (0). A composite score was computed for each of the three categories. Students responded to the test before and after implementation of the program.

Cognitive assignments. Students’ knowledge about how to set up a PA plan was estimated based on their performance on cognitive assignments. More specifically, students were asked to set up a PA plan in writing three times during the implementation of the program. In the seventh session, they were asked to set up a plan for increasing the strength of their arms, abdominal and dorsal muscles. In the twelfth session, they were asked to set up a plan for increasing their aerobic capacity and the strength of their arms, abdominal and dorsal muscles. Finally, they were asked to set up a plan for increasing their overall physical fitness in the fifteenth session. Two trained researchers graded the students’ PA plans independently. A training period preceded the main grading process. Both researchers graded the same workbooks, and their inter-rater agreement was checked. The grading process begins when researchers exceed the threshold of 80% agreement. The grading process lasted two months. Researchers’ inter-rated agreement was also examined during the main grading process. Students’ PA plans were graded based on whether they used specific principles of goal-setting theory. More specifically, four scores were attributed to each PA plan corresponding to the four principles of goal setting (specific, measurable, activity-related and realistic). The time-bound principle was not applied to this specific study because the structure of the program determined the period that students should design a PA plan and achieve their goals.

For example, a PA plan for improving arm strength including the actions of “I will do three (3) sets of 15 push-ups, after doing my homework, in order to increase my arms strength from 17 to 20 repetitions” was considered specific, measurable, activity-related and realistic. Thus, each of the four principles was graded with one (1), and this plan received an overall score of four (4). By contrast, the PA plan “I will do push-ups after doing my homework in order to increase my arms strength” was graded as correct (1) only for specific and relevant principles, while it was graded as false (0) for measurable and realistic principles of goal setting, thus receiving an overall score of two (2).

Retrospective pre–post questionnaire. Based on the assumption that students’ understanding of their level of functioning in PA-related goal setting would have been increased after their participation in the program, a related retrospective measure was used. The retrospective measure was employed in order to record the referential changes in the perceptions of the participants regarding their goal-setting skills, and it was based on the “then–post ratings” method of Howard [55]. Howard [55] provided evidence that regarding the evaluation of various training interventions, the use of pre–post design with self-reports is susceptible to a “response-shift bias”. He defined a response shift as “…a treatment produced a change in a subject’s awareness or understanding of the variable being measured” (p. 320). To provide an example, a student may feel at pretest that they are “average” in goal-setting skills. A subsequent intervention changes their understanding of the behaviors related to goal setting, and during the intervention, they realize that their level of goal-setting skills was below average at pretest. If they improved their level of goal setting due to the intervention from “below average” to “average”, then their pre and post ratings of responsibility would be average, and thus, their improvement would not be evident in the ratings due to the response shift.

Thus, five sets of two items were developed and answered by the students once after the completion of the program. The five sets referred to students’ perceptions regarding knowledge to (a) design a PA plan and implement it, (b) set goals for improving physical fitness, (c) develop a plan to achieve their goals, (d) keep a PA diary and (e) choose activities suitable for strengthening specific body parts. The first item of each set referred to the present (now—denoting the time after the completion of the program) and the second referred to the past (before—denoting the time before the implementation of the program). An example of a two-item set is: “Now, I know how to set PA goals” and “Before attending the program I knew how to set PA goals”. Thus, students answered the same question twice. The first time they answered for the present (how they feel or act now, after receiving the program), and the second time they answered for the past (how they felt or acted before receiving the program). Each item was answered on a 5-point Likert-type response scale, ranging from one (Totally Disagree) to five (Totally Agree).

### 2.4. Data Analysis

Exploratory factor analysis (EFA) was performed in order to examine the construct validity of the retrospective pre–post questionnaire. Cronbach’s alpha reliability coefficients were estimated for all items. Differences in the three composite scores of the multiple-choice test before and after the program were examined by means of repeated-measures Multivariate Analysis of Variance (MANOVA) with two levels (pre–post). A repeated-measures ANOVA was conducted to examine the differences in students’ plans scoring. Finally, students’ answers on the retrospective questionnaire were examined with a repeated-measures MANOVA with five dependent variables (the scores of the five items) and two levels (before the program–after the program).

## 3. Results

### 3.1. Construct Validity and Reliability of Retrospective Pre–Post Questionnaire

A principal component analysis with direct oblique rotation was performed. The analysis revealed two factors explaining a total of 51.45% of the variance for the entire set of variables. The first factor was labeled as a pre-test retrospective questionnaire and explained 35.00% of the variance. The second factor derived was labeled as a post-test retrospective questionnaire and explained 16.45% of the variance. All items in both factors had high loadings (more than 0.62). Cronbach’s alpha coefficient for the pre-test retrospective questionnaire was 0.81, indicating high internal consistency. In contrast, Cronbach’s alpha coefficient for the post-test retrospective questionnaire was 0.62. indicating an acceptable internal consistency.

### 3.2. Differences in Students’ Multiple-Choice Test between the Baseline and Final Test

The repeated-measures MANOVA results showed a significant multivariate effect for students’ knowledge, *F*(3, 240) = 47.10, *p* < 0.001, *η_p_*^2^ = 0.37. Univariate within-group analyses indicated that students’ knowledge significantly increased after the program. Students’ knowledge related to health benefits of PA significantly increased after their participation in the program, *F*(1, 242) = 52.37, *p* < 0.001, *η_p_*^2^ = 0.18. Similarly, students’ awareness of goal setting increased after their participation in the program, *F*(1, 242) = 70.76, *p* < 0.001, *η_p_*^2^ = 0.23. Finally, students’ knowledge related to appropriate activities for exercising specific parts of the human body improved after their participation in the program, *F*(1, 242) = 68.01, *p* < 0.001, *η_p_*^2^ = 0.22. Descriptive statistics regarding the multiple-choice test are presented in Table 1.

### 3.3. Differences in Students’ Knowledge for Setting a Goal-Setting Plan

Mauchly’s test indicated that the assumption of sphericity had been violated, χ^2^ (2) = 14.29, *p* < 0.01. Therefore, degrees of freedom were corrected using Huynh–Feldt estimates (ε = 0.95). The results showed that students’ knowledge in designing a PA plan was significantly affected by the delivery of the program, *F*(1.85, 397.21) = 22.13, *p* < 0.001, *η_p_^2^* = 0.09. Bonferroni post hoc tests showed that participants’ knowledge increased significantly between the first (*M* = 3.07, *SD* = 0.055) and the second plan of PA (*M* = 3.22, *SD* = 0.045). However, there was no significant improvement in students’ knowledge between the second and the third PA plan.

### 3.4. Differences in Students’ Perceptions of Their Knowledge

Repeated-measures MANOVA results showed a significant multivariate effect for students’ perception for their knowledge to design a PA plan, set goals, keep a PA diary and choose *PA* that are most effective for targeting specific body parts, *F*(5, 239) = 88.15, *p* < 0.001, *η_p_*^2^ = 0.64. Univariate within-group analyses indicated that students’ perceptions of their knowledge increased significantly after their participation in the program. Students’ knowledge in designing and implementing a personal PA plan, *F*(1, 243) = 211.56, *p* < 0.001, *η_p_*^2^ = 0.47, significantly increased. Similarly, students’ knowledge in setting a personal plan for achieving goals, *F*(1, 243) = 158.10, *p* < 0.001, *η_p_*^2^ = 0.39, and keeping a PA diary, *F*(1, 243) = 261.92, *p* < 0.001, *η_p_*^2^ = 0.52, significantly increased. Finally, students’ knowledge in choosing activities suitable for strengthening specific body parts, *F*(1, 243) = 196.82, *p* < 0.001, *η_p_*^2^ = 0.45, and setting up a plan to achieve their goals, *F*(1, 243) = 258.42, *p* < 0.001, *η_p_*^2^ = 0.52, also increased significantly.

## 4. Discussion

The purpose of the present study was to examine whether students’ knowledge increased during the implementation of the “After School Exercise” program. The results of the baseline multiple-choice test indicated that the students of the present study had inadequate knowledge of health-related concepts. This is aligned with respective findings of previous studies [34,35]. However, the findings of the post-test suggested that students’ knowledge regarding health-related concepts increased. Further, students’ perceptions regarding their knowledge were aligned with the multiple-choice test results.

Although a significant number of studies have used the goal-setting theory in the PE context (e.g., [46,47] a limited number of them examined whether participants learned to set goals effectively. The results of these studies [48,49,50] are aligned with the results of the present study. However, researchers [30] suggested that the use of the knowledge tests depicts a snapshot of students’ knowledge. By contrast, they argued that the examination of students’ knowledge through the analysis of their cognitive assignments better reflects the learning process and their knowledge development. The present study adopted this suggestion and examined the development of students’ knowledge of designing a PA plan through their assignments. This provided the opportunity to capture not a snapshot of students’ knowledge of goal setting but their ability to effectively convert the theory into practice. Additionally, it must be stated that the cognitive assignments included in the present study demanded of students to apply S.M.A.R.T. goals in order to set up PA plans for improving specific qualities of PA (strength, stamina, etc.). Thus, it can be said that these cognitive assignments help researchers to examine students’ ability to apply not only their knowledge related to S.M.A.R.T. goals but their knowledge related to the suitable activities for exercising specific parts of the human body and specific qualities of PA. Students’ answers to the cognitive assignments revealed that there was an improvement in their knowledge between the first and the second PA plan, but not between the second and the third PA plan. This finding confirmed Vosniadou’s suggestion [21] that learning is a slow, gradual and longitudinal process. However, it can be argued that the incorporation of the cognitive assignments facilitated students’ knowledge and contributed to scoring higher in the post knowledge test. This finding is aligned with the findings of previous studies [17,30,33,37] which showed that cognitive assignments can help students to improve their knowledge. Thus, a program that includes the delivery of health-related knowledge combined with fitness activities and cognitive assignments can lead students to meaningful learning by converting theory into practice and connecting their existing knowledge with new information.

Finally, the findings of the present study are aligned with the findings of previous studies [30,31,35,36,37] which indicated that students’ health-related knowledge increased after they participated in a program. A rational explanation for this could be the fact that students in the PE context can recall, apply and/or experience the information that they received during the lesson [56]. The findings of the present study also confirmed Ennis’s [57] suggestion that a meaningful and coherent PE curriculum may attract students’ attention and help them to gain the skills and knowledge that they can apply to their everyday life. It can be argued that the “After School Exercise” program may help students to “learn to move and learning through movement” [16] which is one of the fundamental PE goals.

Additionally, it can be argued that one of the objectives of the program, which was to raise students’ awareness regarding the benefits of PA, was achieved. Ennis [23] argued that students’ knowledge related to PA and fitness could influence their behavior toward PA. The findings of previous studies [31,33] confirmed that students’ health-related knowledge effectively contributes to their participation in after school PA. This study did not examine to the extent to which students of the present study increase their participation in PA. However, students’ reports in a pilot implementation of the program [58] showed that their participation in the “After School Exercise” program increased the time they spent on PA.

## 5. Limitations

A limitation of the present study was the lack of a control group that could strengthen the findings of the study. A study that includes a control group is likely to allow researchers to examine to the extent to which changes observed in the students of the experimental group are due to the influence of the program, rather than to other factors. Another significant limitation was the fact that the factorial validity and reliability of the multiple-choice test was not examined.

## 6. Conclusions

The findings of the present study indicated that the “After School Exercise” program promoted students’ knowledge development of health-related concepts. This finding confirmed Ennis’s [17] suggestion that a PE program that includes health-related knowledge combined with cognitive assignments can be beneficial for their learning development. It can also be supported that a PE program that facilitates implementation of the gained knowledge to the PE context may affect students’ behavior toward PA and urge them to participate in after school PA. Hence, policymakers and curriculum developers aimed at increasing students’ knowledge and participation in after school PA should adopt and design similar curricula. Future studies should examine the extent to which students’ health-related knowledge influences their participation in after school PA.

## Figures and Tables

**Table 1 children-08-00248-t001:** Means and standard deviation for students’ answers in the multiple-choice test.

Questions	Pre-Test		Post-Test	
	M	SD	M	SD
1.knowledge related to the health benefits of PA	1.96	0.53	2.42	0.50
2.knowledge related to goal-setting	1.92	0.51	2.40	0.44
3.knowledge related to choose activities suitable for strengthening specific body parts	2.36	0.52	2.78	0.31

## Data Availability

The data presented in this study are available on request from the corresponding author.

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
