# Peer review of "Elementary Students’ Knowledge Development during the Implementation of “After School Exercise” Program"

_children, 2021, doi:10.3390/children8030248_

Round 1

Reviewer 1 Report

Children-1133676 presents the results of a one-group, pre-post, intervention to increase student knowledge of the health benefits of physical activity.  The authors report positive results with regards to increasing PA-related knowledge.  The paper is also clear, comprehensive, and well-written.  However, the quality of this research design is limited.  Some aspects of the methodology, or explanation of the methodology can be improved with revision, but the basic design to test the intervention allows for only weak indications of effect.

  1. My primary concern with the design is the lack of a control group. There is nothing that can be done about this issue now, but conclusions should be tempered in that effects cannot be identified without some control condition.
  2. 15 PE teachers accepted and participated. How many were invited?  What percent does this represent?
  3. Program – There does not appear to be any information about adherence of the 16 session program. Do you have data at the teacher or student level about engagement in the program?
  4. My second biggest concern is the instruments used to measure change over the intervention. They do not appear to be particularly reliable or objective.
    1. Multiple choice test – In addition to the content validity, more information on the reliability of the nine question scale is needed. With such a simple measure of knowledge, I am concerned that the intervention was “teaching to the test” and that post-test outcomes may not fully reflect increased knowledge.
    2. Cognitive assessments – The three cognitive assessments were graded by two trained researchers. More information on the inter-rater reliability of the coding would improve the quality of this assessment.
    3. Retrospective-questionnaire – How was the actual questionnaire developed? Validity or reliability evidence? 
  5. I recommend putting greater emphasis on your effect sizes. While I remain suspicious of the improvements reported, due both to design and instrumentation, the effect sizes reported suggest that meaningful change on the instruments occurred during the program.
  6. While the program appears to be effective in increasing health knowledge, it remains unclear if that increase in knowledge would be associated with greater engagement in PA. That could have been tested concurrently in the program.  Even without that component, a greater discussion of how increased knowledge could lead to increased PA is warranted in the discussion.  In my option, the discussion is currently very brief and could be further unpacked.
  7. I recommend checking all reference numbers in the text to the reference list. For example, Howard is listed as 54 in line 209, but appears as 55 in the reference list.

Reviewer 2 Report

The manuscript is well-written and clearly presented. Authors aimed to test whether students who participated in the “After School Exercise” program increased their knowledge regarding goal-setting and the health benefits of PA. I believe it has the potential to make a contribution to the literature; please kindly see the following comments for details. The purpose in the abstract is presented as a long sentence. I believe it could be divided at into least two sentences. Additional information is needed about the program population and curriculum to help readers to understand the findings relative to the study context. Besides, more information is needed about the context of After School Exercise in Greece. Discussion section seems too thin to include the essential information following the results. A more in-depth reflection about what the findings mean may be needed. What are the practical implications or this study? Are there new suggestions for integrating new strategy into “After School Exercise” programs? Finally, I would suggest the authors create a section designated “limitations”. For example, the absence of control group would be one of the major limitations.

Round 2

Reviewer 2 Report

The authors’ revision appears to have addressed my concerns. I believe this paper has the potential to make a contribution to the literature. Good luck with the submission process!